# Design, Synthesis and Molecular Docking Analysis of Flavonoid Derivatives as Potential Telomerase Inhibitors

**DOI:** 10.3390/molecules24173180

**Published:** 2019-09-01

**Authors:** Zhan-Fang Fan, Sai-Tim Ho, Rui Wen, Ya Fu, Lei Zhang, Jian Wang, Chun Hu, Pang-Chui Shaw, Yang Liu, Mao-Sheng Cheng

**Affiliations:** 1Key Laboratory of Structure-Based Drugs Design and Discovery (Ministry of Education), School of Pharmaceutical Engineering, Shenyang Pharmaceutical University, Shenyang 110016, China; 2School of Life Sciences, The Chinese University of Hong Kong, Shatin, Hong Kong, China

**Keywords:** human telomerase holoenzyme, flavonoid, antiproliferative activity, molecular modeling

## Abstract

Based on the structural scaffolds of natural products, two series of flavonoid derivatives, for a total of twelve compounds, were designed and synthesized as potential human telomerase inhibitors. Using a modified TRAP-PCR assay, compound **5c** exhibited the most potent inhibitory activity against human telomerase with an IC_50_ value of less than 50 μM. In vitro, the results demonstrated that compound **5c** had potent anticancer activity against five classes of tumor cell lines. The molecular docking and molecular dynamics analyses binding to the human telomerase holoenzyme were performed to elucidate the binding mode of active compound **5c**. This finding helps the rational design of more potent telomerase inhibitors based on the structural scaffolds of natural products.

## 1. Introduction

Cancer is one of the leading causes of mortality worldwide. Approximately one-third of people are affected by cancer during their lives [1]. In 2018, the American Cancer Society published the global cancer statistics 2018, which estimated that there will be about 18.1 million new cancer cases and 9.6 million cancer deaths worldwide [2]. Lots of studies have indicated that telomerase activity can be detected in approximately 80–90% cancer cells; however, in somatic cells, this enzyme is relatively low [3,4]. As the community of destiny of telomere, telomerase ensures the indefinite proliferation of cancer cells and has been chosen as a hot target for drug development in cancer therapy [5].

Multiple telomerase inhibitors have been produced, including natural and synthetic products and modified oligonucleotides, through targeting of the catalytic core [5]. In addition, as telomere binding agents, quadruplex ligands (G4) also play a significant role in inhibiting the activity of telomerase [6,7,8]. Due to the side effects of synthetic products, such as multidrug resistance, toxicity, and poor bioavailability, it is preferred that telomerase inhibitors be isolated from natural plant and marine materials, including secondary metabolites such as polyphenols, alkaloids, terpenoids, xanthones, and sesquiterpenes [9,10,11]. Among them, flavonoids, as naturally occurring polyphenols, are widely distributed in many field plants and exhibit multitudinous biological and pharmacological properties, for instance, antitumor, cytotoxic, anti-inflammatory, antioxidant, cardiovascular and antidiabetic effects. Several studies have demonstrated that flavonoids can treat and prevent cancer by inhibiting telomerase activity and inducing apoptosis [12,13,14,15]. However, few flavonoids compounds have been developed into commercially available drugs due to their relatively weak anticancer potency and the uncertain mechanism of action. To date, there is no inhibitor based on the flavonoid skeleton that has been approved by the FDA. Therefore, to explore novel and potent telomerase inhibitors based on the structural scaffolds of natural product flavonoids that have enhanced anticancer activity and can reveal the particular mechanisms remains a challenging task.

As a class of polyhydroxy products, carbohydrates are widespread and ubiquitous in nature and have excellent water solubility and various bioactivities, which have made them a reliable and valuable source for drug design [16]. In addition, the “click reaction”, namely the Cu(I)-catalyzed azide-alkyne 1,3-dipolar cycloaddition, has been widely used to construct the 1,2,3-triazoles fragment, which can play an important role in drug design [17]. Moreover, numerous structures and drugs bearing a cinnamoyl moiety have shown strong inhibition against various tumor cell lines [18,19,20,21]. Previously, based on natural product design, our group reported a series of triterpenoid saponin derivatives containing a triazole linkage and *N*-acetylglucosamine derivatives substituted with cinnamoyl groups and they all had remarkable antitumor activity both in vitro and in vivo [22,23,24,25]. Inspired by these previous results, two series of novel flavonoid derivatives with glycosyl fragments linked by a triazole and containing an *N*-acetylglucosamine substituted with a cinnamoyl group were designed and synthesized to enhance inhibitory ability and elucidate their pharmacological mechanism (Figure 1).

In this study, we synthesized two series of flavonoid derivatives, for a total of twelve compounds according to the split principle. Telomerase inhibition activity in vitro and antiproliferative activity against five cancer cells lines were assayed to evaluate the possibility of these compounds as potential telomerase inhibitors. Compound **5c** showed the most potent activities both in the telomerase inhibition assay and against all tested cell lines.

To date, for the human telomerase holoenzyme, there is no cocrystallization of protein and ligand complexes available [26]. In this report, the active site of the protein structure was confirmed via alignment with a similar protein discovered from *Tribolium castaneum* (PDB:5CQG), whose structure is in complex with the highly specific inhibitor BIBR1532 [27]. Then, molecular docking simulations were performed to elucidate the binding mode of active compound **5c**.

## 2. Results and Discussion

### 2.1. Chemistry

Six flavonoid derivatives bearing glycosyl and triazole moieties were synthesized. The general synthetic strategy for the formation of the target compounds is shown in Scheme 1. Kaempferol (**1**), as the starting material, was protected by acetylation to give compound **2**. Compound **3** was synthesized via selective deprotection with compound **2** and then allylation to obtain compound **4**. Three azidosaccharide donors **D_1_**–**D_3_** were prepared by the azidation of compounds **C_1_**–**C_3_**, which were obtained by bromination of **B_1_**–**B_3_** with HBr-AcOH. The cyclization of **4** with **D_1_**–**D_3_** through the classic “click-reaction” produced compounds **5a**–**5c**, which were per deacetylated with a catalytic amount of NaOMe in CH_3_OH to give compounds **6a**–**6c**.

The second series of flavonoid derivatives containing *N*-acetylglucosamine substituted by a cinnamoyl group were synthesized, and the general synthetic route is shown in Scheme 2. Here, compound **7** could be efficiently obtained from intermediate **3** via benzyl protection, followed by deacetylation to give compound **8**. Subsequently, treatment of **8** with glycosyl bromide [16] in the presence of molecular sieve led to the formation of **9**, followed by acetylation to give **10**. Then, to a solution of compound **10** in acetic acid, zinc powder was added in batches to selectively remove the Troc group to liberate **11**. Then, compound **12** was generated by the debenzylation of **11** catalyzed with H_2_-Pd/C, followed by condensation with substituted cinnamoyl chloride to give compounds **13a**–**13f**. Compounds **14a**–**14f** were obtained by deacetylation of compounds **13a**–**13f** with 1 mol/L NaOCH_3_ in CH_3_OH**.**

### 2.2. Biological Activity

#### 2.2.1. Telomerase Inhibitory Assay

According to previous reports, the exposed hydroxy group of the flavonoid is a functional group that enhances its anticancer potential [5,13,14]. Herein, compounds **6a**–**6c**, **14a**–**14f** were screened for their in vitro telomerase inhibitory activity against HeLa human cervical cancer cells via the telomere repeat amplification protocol–polymerase chain reaction (TRAP–PCR) assay, and compound **5c** was chosen as a control. The results are shown in Figure 2. Unexpectedly, among them, compounds **5c** and **14e** showed potent inhibitory activity against telomerase compared with baicalin as a reference. Surprisingly, compound **5c** showed the most effective activity against telomerase with IC_50_ < 50 μM (baicalin: IC_50_ > 100 μM). The structure–activity relationship (SAR) of these flavonoid derivatives demonstrated that compounds with acetyl protection displayed potential inhibitory activity.

#### 2.2.2. Antiproliferation Assay

To further explore the structure–activity relationship (SAR) of these compounds and discover the most potent compound, four flavonoid derivatives, **5c**, **6c**, **14b** and **14e** were evaluated for their antiproliferative activity against human lung cancer cells (A549), human hepatoma cells (HepG2), human cervical cancer cells (HeLa), human gastric cancer cells (MGC-803), and human gastric cancer cells (SGC-7901) by comparison with 5-fluorouracil. The results are summarized in Figure 3 and Table 1.

As shown in Figure 3, these results suggested that compound **5c** was an excellent anticancer agent with potential broad-spectrum anticancer activity compared with the positive control 5-fluorouracil. The anticancer activity of compound **5c** was consistent with the SAR of the telomerase inhibitory activity. It has been demonstrated that the potent anticancer activity of the synthetic compound **5c** is correlated with its telomerase inhibitory activity.

Next, compound **5c** was estimated on two normal human cell lines (Hacat and BEAS-2B). As shown in Table 2, the IC_50_ values of compound **5c** against the two normal human cells were greater than 200 μM, which proves that compound **5c** owns its selective effect.

### 2.3. Molecular Simulation Analyses

In 2018, the Kathleen Collins group first presented the cryo-electron microscopy structure of the substrate-bound human telomerase holoenzyme, as well as the PDB coordinates [25]. However, there is no cocrystallization of protein and ligand complexes available. In this report, to evaluate the binding mode of active compound **5c** compared with the inactive compound **6c**, the active site of the protein structure was confirmed via alignment with a similar protein discovered from *T. castaneum* (PDB: 5CQG), whose structure is in complex with the highly specific inhibitor BIBR1532. Then, the docked complexes (**5c**, **6c**, and BIBR1532) were chosen and submitted to perform a 100 ns molecular dynamics (MD) simulation. The dynamic properties of the three complexes were then subjected to analysis of the trajectory data obtained from the 100 ns MD simulations. The root-mean-square deviation (RMSD), root-mean-square fluctuation (RMSF), and protein–ligand contacts were used to evaluate each system in the molecular dynamics studies, as shown in Figure 4, Figure 5 and Figure 6, respectively. Meanwhile, the superimposition of **5c** and **6c** with BIBR1532 in the active site of telomerase is presented in Figure 7.

As shown in Figure 4, during the simulation, the RMSD value of the BIBR1532/telomerase complex configuration kept increasing until it stabilized at approximately 6.3 Å after 50 ns. Similarly, the RMSD value of the **5c**/telomerase complex finally reached an RMSD plateau at approximately 6.2 Å after 60 ns, although the complex underwent small fluctuations during the periods between 30−50 ns. These results indicated that the BIBR1532/telomerase and **5c**/telomerase complexes reached a steady state at the end of the MD simulations. Regrettably, the RMSD value of the **6c**/telomerase complex underwent fluctuations by a large margin until it stabilized at approximately 8.2 Å after 90 ns. The root-mean-square fluctuation (RMSF) plots for BIBR1532 and compounds **5c** and **6c** can explain the local changes in the protein chain during the MD simulations, which are shown in Figure 5. It was obvious that the overall trend of the polylines was substantially similar between BIBR1532 and **5c**, indicating that the compound **5c** and BIBR1532 had similar stability and mode of interaction.

As shown in Figure 6, four types of protein−ligand interactions were monitored throughout the MD simulations: hydrogen bonds, hydrophobic interactions, ionic interactions, and water bridges. Analysis of the BIBR1532/telomerase complex trajectory data showed the existence of hydrogen bonds between Asn 421, Arg 433, Lys 437 of telomerase and the carbonyl and carboxylic acid groups of BIBR1532. The hydrogen bonds in which Arg 433 participated accounted for 117% of the entire MD trajectory, indicating that this residue formed more than one hydrogen bond. In addition, the hydrogen bonds formed by Asn 421 and Lys 437 were found to account for 20% and 21% of the entire MD trajectory. From the above analysis, we found that the hydrogen-bonding interactions between the carbonyl group and Asn 421, Arg 433 and Lys 437 played significant roles in ligand binding. For hydrophobic contacts, the interactions with Ala 438 were relatively stable and were observed in 72% of the MD simulations. Several weaker ionic contacts were also found in this complex with Arg 433 and Lys 437 (Figure 6A). From the trajectory analysis of the **5c**/telomerase complex, we can see that the kaempferol acetyl moiety of **5c** can mimic the benzoic acid moiety of BIBR1532 and form the key hydrogen bonds with Asn 421 and Lys 437, which accounts for 147% and 21%, respectively. Meanwhile, Lys 437 and Phe 434 produced hydrophobic interactions with compound **5c**, accounting for 48% and 32%, respectively (Figure 6B). From the trajectory analysis of the **6c**/telomerase complex, it was not surprising to find that the binding mode changed substantially. Here, it was not hard to see that the key binding interaction was hydrogen bonds between Lys 418 and Asn 424, which has much imparity between the above results (Figure 6C).

With deep analysis of the MD simulation results of **5c** and **6c**, compared with BIBR1532, the interactions between **5c** and the amino acid residues of telomerase were similar to those of BIBR1532. Conversely, **6c** showed substantial differences as discussed above. All of the above verified that the docking results were accurate (Figure 7 and Figure 8) and all the results were consistent with the active in vitro data.

## 3. Materials and Methods

### 3.1. Chemistry

Reagents (Energy Chemical, Shanghai, China) were used without further purification unless otherwise specified. Solvents were dried and redistilled prior to use in the usual way. Analytical TLC was performed using silica gel HF_254_ (Qingdao Haiyang Chemical, Qingdao, Shandong Province, China). Preparative column chromatography was performed using silica gel H. Melting points were obtained on a Büchi melting point B-540 apparatus. NMR spectra were recorded on a Bruker ARX 600 MHz spectrometer (Bruker, Zurich, Zürich, Swiss) with TMS as the internal standard. NMR spectra were analyzed and interpreted using MestreNova (Mestrelab Research, Spain). HRMS were obtained on a Bruker microTOF_Q spectrometer (Agilent, Santa Clara, CA, USA). The gel was stained with SYBR Green I Nuclei A (Invitrogen, Carlsbad, CA, USA) according to the manufacturer’s instructions. A Telomerase PCR ELISA kit (Roche, Palo Alto, CA, USA) was used per the manufacturer’s instructions.

#### 3.1.1. General Procedure for the Preparation of Compounds **5a**–**5c**

To a solution of compound **4** (1 g, 2.22 mmol) and compounds **D_1_**–**D_3_** (2.22 mmol) in 20 mL of acetonitrile, cuprous iodide (0.845 g, 4.44 mmol) and triethylamine (0.925 mL, 6.66 mmol) were added and the reaction mixture was stirred at room temperature for 1 h. The mixture was quenched by the addition of saturated ammonium chloride solution, followed by extraction with ethyl acetate. The organic layer was washed with brine, dried over anhydrous Na_2_SO_4_, filtered, and the solvent was removed in vacuo. The residue was purified by silica gel column chromatography (petroleum ether: ethyl acetate = 2:1) to afford compounds **5a**–**5c**, the NMR spectrums of compounds **5a**–**5c** were contained in Appendix A.

*7-O-((1-(2,3,4,6-tetra-O-acetyl-β-d-glucopyranosyl)-1H-1,2,3-triazol-4-yl)-methyl)-3,5,4′-tri-O-acetyl-kaempferol* (**5a**): Yellow solid, 87% yield, m.p. 173.5–175.2 °C; ^1^H-NMR (600 MHz, DMSO-d_6_) δ 8.64 (s, 1H), 7.95 (d, *J* = 8.5 Hz, 2H), 7.43 (d, *J* = 2.4 Hz, 1H), 7.40 (d, *J* = 8.5 Hz, 2H), 6.93 (d, *J* = 2.4 Hz, 1H), 6.39 (d, *J* = 9.1 Hz, 1H), 5.68 (t, *J* = 9.3 Hz, 1H), 5.56 (t, *J* = 9.5 Hz, 1H), 5.39 (d, *J* = 2.6 Hz, 2H), 5.19 (t, *J* = 9.7 Hz, 1H), 4.38 (ddd, *J* = 10.2, 5.3, 2.3 Hz, 1H), 4.16–4.07 (m, 2H), 2.32 (d, *J* = 4.6 Hz, 9H), 2.03 (s, 3H), 1.99 (s, 3H), 1.96 (s, 3H), 1.76 (s, 3H). HRMS (ESI): calcd. for [M + H]^+^ C_38_H_38_N_3_O_18_^+^: 824.2150, found 824.2159.

*7-O-((1-(2,3,4,6-tetra-O-acetyl-β-d-galactopyranosyl)-1H-1,2,3-triazol-4-yl)-methyl)-3,5,4′-tri-O-acetyl-kaempferol* (**5b**): Yellow solid, 87% yield, m.p. 173.7–175.8 °C; ^1^H-NMR (600 MHz, DMSO-d_6_) δ 8.63 (s, 1H), 7.95 (d, *J* = 8.8 Hz, 2H), 7.43 (d, *J* = 2.5Hz, 1H), 7.40 (d, *J* = 8.8 Hz, 2H), 6.92 (d, *J* = 2.5 Hz, 1H), 6.41 (d, *J* = 9.5 Hz, 1H), 5.69 (t, *J* = 9.5 Hz, 1H), 5.57 (t, *J* = 9.7 Hz, 1H), 5.39 (d, *J* = 2.6 Hz, 2H), 5.19 (t, *J* = 9.7 Hz, 1H), 4.38 (dd, *J* = 4.4, 2.3 Hz, 1H), 4.09 (m, 2H), 2.32 (d, *J* = 4.6 Hz, 9H), 2.03 (s, 3H), 1.99 (s, 3H), 1.96 (s, 3H), 1.76 (s, 3H). HRMS (ESI): calcd. for [M + H]^+^ C_38_H_38_N_3_O_18_^+^: 824.2150, found 824.2201.

*7-O-((1-(2,3,4-tri-O-acetyl-a-L-arabinopyranosyl)-1H-1,2,3-triazol-4-yl)-methyl)-3,5,4′-tri-O-acetyl-kaempferol* (**5c**): Yellow solid, 85% yield, m.p. 175.8–176.2 °C; ^1^H-NMR (600 MHz, DMSO-d_6_) δ 8.55 (s, 1H), 8.00–7.91 (m, 2H), 7.45 (d, *J* = 2.4 Hz, 1H), 7.41–7.37 (m, 2H), 6.93 (d, *J* = 2.4 Hz, 1H), 6.18 (d, *J* = 9.2 Hz, 1H), 5.60 (t, *J* = 9.6 Hz, 1H), 5.41 (dd, *J* = 10.2, 3.6 Hz, 1H), 5.37 (s, 2H), 5.32 (dt, *J* = 3.4, 1.5 Hz, 1H), 4.21–4.16 (m, 1H), 4.05 (dd, *J* = 13.2, 1.9 Hz, 1H), 2.39–2.24 (m, 9H), 2.16 (s, 3H), 1.95 (s, 3H), 1.78 (s, 3H). HRMS (ESI): calcd. for [M + H]^+^ C_35_H_34_N_3_O_16_^+^: 752.1939, found 752.1953.

#### 3.1.2. General Procedure for the Preparation of Compounds **6a**–**6c**

To a solution of compounds **5a**–**5c** (1 g) in 30 mL of DCM-CH_3_OH (1:1), freshly prepared NaOMe in MeOH solution (1.0 mol/L, 2.5 eq.) was added dropwise with stirring. After 0.5 h, the mixture was neutralized with Dowex H^+^ resin to pH 7 and then filtered. The filtrate was concentrated and purified by silica gel column chromatography (DCM:MeOH = 8:1) to afford compounds **6a**–**6c**, the NMR spectrums of compounds **6a**–**6c** were contained in Appendix A.

*7-O-((1-(β-d-glucopyranosyl)-1H-1,2,3-triazol-4-yl)-methyl)-kaempferol* (**6a**): Yellow solid, 45% yield, m.p. 175.8–176.2 °C; ^1^H-NMR (600 MHz, DMSO-d_6_) δ 12.49 (s, 1H), 10.16 (s, 1H), 9.51 (s, 1H), 8.49 (s, 1H), 8.08 (d, *J* = 8.9 Hz, 2H), 6.95 (d, *J* = 8.9 Hz, 2H), 6.92 (d, *J* = 2.2 Hz, 1H), 6.45 (d, *J* = 2.2 Hz, 1H), 5.57 (d, *J* = 9.2 Hz, 1H), 5.42 (d, *J* = 5.9 Hz, 1H), 5.29 (s, 1H), 5.28 (s, 2H), 5.16 (d, *J* = 5.2 Hz, 1H), 4.62 (t, *J* = 5.5 Hz, 1H), 3.78 (td, *J* = 9.1, 5.8 Hz, 1H), 3.69 (dd, *J* = 9.9, 5.7 Hz, 1H), 3.49–3.35 (m,4H). HRMS (ESI): calcd. for [M + H]^+^ C_24_H_24_N_3_O_11_^+^: 530.1411, found 530.1423.

*7-O-((1-(β-d-galactopyranosyl)-1H-1,2,3-triazol-4-yl)-methyl)-kaempferol* (**6b**): Yellow solid, 47% yield, m.p. 175.8–176.2 °C; ^1^H-NMR (600 MHz, DMSO-d_6_) δ 12.49 (s, 1H), 10.13 (s, 1H), 9.51 (s, 1H), 8.43 (s, 1H), 8.09 (d, *J* = 8.5 Hz, 2H), 6.95 (d, *J* = 8.9 Hz, 2H), 6.92 (d, *J* = 2.8 Hz, 1H), 6.46 (d, *J* = 2.2 Hz, 1H), 5.52 (d, *J* = 9.1 Hz, 1H), 5.29 (s, 2H), 5.26 (d, *J* = 7.6 Hz, 1H), 5.02 (d, *J* = 5.7 Hz, 1H), 4.06 (td, *J* = 9.3, 5.9 Hz, 1H), 3.75 (dt, *J* = 20.0, 5.1 Hz, 2H), 3.59–3.43 (m,3H). HRMS (ESI): calcd. for [M + H]^+^ C_24_H_24_N_3_O_11_^+^: 530.1411, found 530.1364.

*7-O-((1-(a-L-arabinopyranosyl)-1H-1,2,3-triazol-4-yl)-methyl)-kaempferol* (**6c**): Yellow solid, 42% yield, m.p. 175.8–176.2 °C; ^1^H-NMR (600 MHz, DMSO-d6) δ 12.48 (s, 1H), 10.12 (s, 1H), 9.51 (s, 1H), 8.41 (s, 1H), 8.08 (d, *J* = 8.9 Hz, 2H), 6.94 (d, *J* = 8.9 Hz, 2H), 6.91 (d, *J* = 2.2 Hz, 1H), 6.45 (d, *J* = 2.2 Hz, 1H), 5.45 (d, *J* = 9.1 Hz, 1H), 5.29 (s, 2H), 5.01 (d, *J* = 5.8 Hz, 1H), 4.79 (d, *J* = 4.5 Hz, 1H), 4.05 (td, *J* = 9.2, 5.8 Hz, 1H), 3.85–3.71 (m, 3H), 3.56 (ddd, *J* = 9.2, 5.6, 3.3 Hz, 1H), 3.16 (d, *J* = 4.7 Hz, 1H). HRMS (ESI): calcd. for [M + H]^+^ C_23_H_22_N_3_O_10_^+^: 500.1305, found 500.1298.

#### 3.1.3. General Procedure for the Preparation of Compounds **14a**–**14f**

To a solution of compound **12** (0.4 g, 0.6 mmol) and pyridine (2.5 mL) in 10 mL dry DCM, substituted cinnamoyl chloride (1.32 mmol) was added dropwise at room temperature, and the reaction mixture stirred for 15 min. The organic layer was washed with water, saturated sodium hydrogen carbonate, and brine, dried over anhydrous Na_2_SO_4_, filtered, and the solvent was removed in vacuo. The residue was concentrated and purified by silica gel column chromatography (petroleum ether: ethyl acetate = 5:1) to obtain compounds **13a**–**13f**. To a solution of **13a**–**13f** in 20 mL DCM-CH_3_OH (1:1), a 1.0 mol/L of NaOMe in MeOH solution (2.5 eq) was added dropwise with stirring. After 0.5 h, the mixture was neutralized with Dowex H^+^ resin to pH 7 and then filtered. The filtrate was removed in vacuo and the residue was purified by silica gel column chromatography (DCM:MeOH = 8:1) to afford compounds **14a**–**14f**, the NMR spectrums of compounds **14a**–**14f** were contained in Appendix A.

*3-O-(2-deoxy-2-((3-(4-fluorophenyl)-1-oxo-2-propenyl)amino)-β-d-glucopyranosyl)-kaempferol* (**14a**): Yellow solid, 45% yield, m.p. 175.8–176.2 °C; ^1^H-NMR (600 MHz, DMSO-d_6_) δ: 12.65 (s, 1H), 11.11 (s, 1H), 10.38 (s,1H), 8.29 (d, *J* = 8.8 Hz, 1H), 8.11 (d, *J* = 8.8 Hz, 2H), 7.62 (dd, *J* = 8.5, 5.8Hz, 2H), 7.41 (d, *J* = 15.7 Hz, 1H), 7.25 (t, *J* = 8.9 Hz, 2H,), 6.92 (d, *J* = 7.5 Hz, 2H), 6.62 (d, *J* = 15.6 Hz, 1H), 6.49 (s, 1H), 6.23 (s, 1H), 5.70 (d, *J* = 8.4 Hz, 1H), 5.16 (d, *J* = 3.1 Hz, 2H), 4.42–4.34 (m, 1H), 3.89 (m, 1H), 3.57 (m, 1H), 3.43 (dd, *J* = 9.9, 6.1 Hz, 2H), 3.16 (d, *J* = 5.2 Hz, 2H). HRMS (ESI): calcd. for [M + H]^+^ C_30_H_27_FNO_11_^+^: 596.1568, found 596.1549.

*3-O-(2-deoxy-2-((3-(4-chlorophenyl)-1-oxo-2-propenyl)amino)-β-d-glucopyranosyl)-kaempferol* (**14b**): Yellow solid, 35% yield, m.p. 198.2–199.7 °C; ^1^H-NMR (600 MHz, DMSO-d_6_) δ 12.66 (s, 1H), 10.93 (s, 1H), 10.25 (s, 1H), 8.28 (d, *J* = 9.2 Hz, 1H), 8.12 (d, *J* = 9.0 Hz, 2H), 7.60 (d, *J* = 8.6 Hz, 2H), 7.48 (d, *J* = 8.6 Hz, 2H), 7.41 (d, *J* = 15.8 Hz, 1H), 6.94–6.89 (m, 2H), 6.68 (d, *J* = 15.8 Hz, 1H), 6.46 (d, *J* = 2.1 Hz, 1H), 6.20 (d, *J* = 2.1 Hz, 1H), 5.71 (d, *J* = 8.4 Hz, 1H), 5.10 (t, *J* = 4.6 Hz, 2H), 4.32 (t, *J* = 5.6 Hz, 1H), 3.95–3.84 (m, 1H), 3.59–3.55 (m, 1H), 3.46–3.42 (m, 1H), 3.36 (s,1H), 3.13 (td, *J* = 6.6, 5.5, 3.5 Hz, 2H). HRMS (ESI): calcd. for [M + H]^+^ C_30_H_27_ClNO_11_^+^: 612.1273, found 612.1306.

*3-O-(2-deoxy-2-((3-(3-fluorophenyl)-1-oxo-2-propenyl)amino)-β-d-glucopyranosyl)-kaempferol* (**14c**): Yellow solid, 30% yield, m.p. 181.8–183.2 °C; ^1^H-NMR (600 MHz, DMSO-d_6_) δ: 12.66 (s, 1H), 10.42 (s, 2H), 8.30 (d, *J* = 9.6 Hz, 1H), 8.11 (d, *J* = 6.9 Hz, 2H), 7.47 (d, *J* = 1.4 Hz, 1H), 7.43 (d, *J* = 4.9 Hz, 2H), 7.40 (s, 1H), 7.21 (d, *J* = 16.3 Hz, 1H), 6.90 (d, *J* = 6.7 Hz,2H), 6.70 (d, *J* = 16.4 Hz, 1H), 6.42 (s, 1H), 6.17 (s, 1H), 5.71 (d, *J* = 8.0 Hz, 1H), 5.07 (d, *J* = 18.5 Hz, 2H), 4.35 (s, 1H), 3.89 (dd, *J* = 18.5, 9.7Hz, 1H), 3.58 (d, *J* = 11.3 Hz, 1H), 3.43 (s, 2H), 3.11 (s, 2H). HRMS (ESI): calcd. for [M − H]^-^ C_30_H_27_FNO_11_^−^: 594.1412, found 594.1418. 

*3-O-(2-deoxy-2-((3-(3-hydroxyphenyl)-1-oxo-2-propenyl)amino)-β-d-glucopyranosyl)-kaempferol* (**14d**): Yellow solid, 35% yield, m.p. 198.2–199.7 °C; ^1^H-NMR (600 MHz, DMSO-d_6_) δ: 12.64 (s, 1H), 11.16 (s, 1H), 10.42 (s,1H), 9.76 (s, 1H), 8.31 (d, *J* = 9.1 Hz, 1H), 8.10 (d, *J* = 8.9 Hz, 2H),7.31 (d, *J* = 15.7 Hz, 1H), 7.19 (t, *J* = 7.8 Hz, 1H), 6.97 (s, 2H), 6.95 (s,1H), 6.92 (s, 1H), 6.80 (d, *J* = 7.6 Hz, 1H), 6.59 (d, *J* = 15.8 Hz, 1H), 6.51 (d, *J* = 1.9 Hz, 1H), 6.25 (d, *J* = 1.9 Hz, 1H), 5.68 (d, *J* = 8.4 Hz, 1H), 5.19 (dd, *J* = 11.8, 5.3 Hz, 2H), 4.39 (t, *J* = 5.6 Hz, 1H), 3.88 (dd, *J* = 18.7, 9.2Hz, 1H), 3.55 (d, *J* = 5.1 Hz, 1H), 3.49–3.41 (m, 2H), 3.10 (s, 2H). HRMS (ESI): calcd. for [M + H]^+^ C_30_H_28_NO_12_^+^: 594.1612, found 594.1576.

*3-O-(2-deoxy-2-((3-(4-trifluoromethylphenyl)-1-oxo-2-propenyl)amino)-β-d-glucopyranosyl)-kaempferol* (**14e**): Yellow solid, 35% yield, m.p. 178.8–180.1 °C; ^1^H-NMR (600 MHz, DMSO-d_6_) δ: 12.66 (s, 1H), 10.28 (s, 2H), 8.31 (d, *J* = 9.2 Hz, 1H), 8.12 (d, *J* = 8.9 Hz, 2H), 7.91 (s, 1H), 7.88 (d, *J* = 7.8 Hz, 1H), 7.73 (d, *J* = 8.1 Hz, 1H), 7.67 (d, *J* = 7.8 Hz, 1H), 7.51 (d, *J* = 15.8 Hz, 1H), 6.90 (d, *J* = 8.9 Hz, 2H), 6.80 (d, *J* = 15.9 Hz, 1H), 6.41 (d, *J* = 1.9 Hz,1H), 6.16 (d, *J* = 1.9 Hz, 1H), 5.71 (d, *J* = 8.4 Hz, 1H), 5.12 (dd, *J* = 11.1, 3.7Hz, 2H), 4.34 (t, *J* = 5.2 Hz, 1H), 3.89 (dd, *J* = 18.6, 9.3 Hz, 1H), 3.58 (dd, *J* = 11.2,4.5 Hz, 1H), 3.45 (m, 2H), 3.12 (s, 2H). HRMS (ESI): calcd. for [M + H]^+^ C_31_H_27_F_3_NO_11_^+^: 646.1536, found 646.1522.

*3-O-(2-deoxy-2-((3-(4-methoxylphenyl)-1-oxo-2-propenyl)amino)-β-d-glucopyranosyl)-kaempferol* (**14f**): Yellow solid, 35% yield, m.p. 177.9–179.6 °C; ^1^H-NMR (600 MHz, DMSO-d_6_) δ: 12.67 (s, 1H), 10.86 (s, 1H), 10.22 (s, 1H), 8.16 (d, *J* = 9.4 Hz, 1H), 8.12 (d, *J* = 8.7 Hz, 2H), 7.51 (d, *J* = 8.5 Hz, 2H), 7.36 (d, *J* = 15.7 Hz, 1H), 6.97 (d, *J* = 8.5 Hz, 2H), 6.90 (d, *J* = 8.7 Hz, 2H), 6.51 (d, *J* = 15.9 Hz, 1H), 6.43 (s, 1H), 6.18 (s, 1H), 5.70 (d, *J* = 8.4 Hz, 1H), 5.09 (d, *J* = 5.9 Hz, 2H), 4.32 (t, *J* = 5.0 Hz, 1H), 3.89 (dd, *J* = 18.5, 9.3 Hz, 1H), 3.78 (s, 3H), 3.58 (dd, *J* = 10.9, 4.8 Hz), 3.45–3.38 (m, 2H), 3.11 (s, 2H). HRMS (ESI): calcd. for [M + H]^+^ C_31_H_30_NO_12_^+^: 608.1768, found 608.1823.

### 3.2. Biological Activity

#### 3.2.1. Telomerase Activity Assays

Compounds **5c**, **6a**–**6c** and **14a**–**14f** were tested to search for inhibitors based on the structural framework of natural products with telomerase activity by using the TRAP-PCR assay. The idea is that if the compound does not inhibit telomerase activity at the labeled concentration, the number and intensity of ladders of the lane should be comparable to the positive control lane (as there is no inhibition, full-strength telomerase within cell lysate will lead to maximum number and intensity of ladders). If there is no lysate added, only the internal control band will be visible. A lane with potent inhibition will appear like the negative control lane. In detail, HeLa cells were first maintained in RPMI 1640 buffer, in which 10% fetal bovine serum was supplemented at 37 °C in a humidified atmosphere containing 5% CO_2_. After trypsinization, 2 × 10^5^ HeLa cells in the logarithmic growth phase were counted with a hemocytometer and collected in a tube, followed by incubation with baicalin and the drugs at a series of concentrations. After 24 h, 200 µL of CHAPS lysis buffer was added to resuspend the cell pellet followed by incubation on ice for 30 min. The cell suspension was centrifuged at 14,000 rpm for 20 min at 4 °C, and the cell lysate (supernatant) was transferred to a new tube. The TRAP reaction was set up on ice in a 0.1 mL PCR tube. The reaction tube was incubated in 2720 Thermocycler (BioTek Instruments, Winooski, VT, USA) (Applied Biosystem) at 30 °C for 30 min to undergo TS extension followed by the 95 °C heat inactivation of telomerase for 5 min. Another reaction mix for the PCR was set up and subsequently added to the reaction tube. PCR was then initiated at 94 °C for 30 s, 60 °C for 30 s, and 72 °C for 30 s. The PCR products were analyzed by 12.5% nondenaturing gel electrophoresis in 0.5X TBE running buffer. The gel was stained with SYBR Green I Nuclei A (Invitrogen, Carlsbad, CA, USA) according to the manufacturer’s instructions, and the results were obtained by autoradiography.

#### 3.2.2. Cell Proliferation Assays

The antiproliferative activity of title compounds **5c**, **6c**, **14b** and **14e** against the five tumor cell lines (A549, HepG2, HeLa, MGC-803 and SGC-7901) as well as compound **5c** against human normal cell lines (Hacat and BEAS-2B) were evaluated using a standard CCK-8-based colorimetric assay. All the above cell lines were provided by Jiangsu Keygen Biotech Co., Ltd. (Nanjing, China). In detail, target tumor cells were grown to the log phase in RPMI 1640 buffer, in which 10% fetal bovine serum was supplemented, and dilution to 5.5 × 10^5^ cells mL^-1^ (A549, 4 × 10^4^ cells mL^−1^) with complete medium. Then each well of 96-well culture plates added 100 μL of the obtained cell suspension and placed in at 37 °C in a 5% CO_2_ atmosphere for 24 h for incubation before being subjected to the antiproliferation assessment. Then, 100 μL of a series of concentrations of drug-containing medium were dispensed into wells to maintain final concentrations of 200, 100, 50, 25, 12.5, 6.25, 3.13, 1.56, and 0.78 μM. Each concentration was tested in triplicate, and 5-fluorouracil (5-FU) (Sigma–Aldrich, St. Louis, USA) was used as the positive control. After 48 h of incubation, cell survival was determined by the addition of 10 μL of CCK-8 (BIOSHARP, 35002) working solution. After postincubation at 37 °C for 3 h, the plates were vortexed for 10 min to remove the air bubbles. The optical absorbance was measured at 450 nm with a microplate reader (BioTek Instruments, Winooski, VT, USA, EL-x800). The data represented the mean of three independent experiments in triplicate and were expressed as mean ± SD. The IC_50_ value was defined as the concentration at which 50% of the cells could survive. The IC_50_ values were calculated by fitting with the three-parameter Hill equation, with SPSS Statistics 19 where y is percent inhibition, x is inhibitor concentration, n is the slope of the concentration–response curve (Hill slope), and V_max_ is maximal inhibition from three independent assays.
$${y=V}_{\max}(\frac{x^{n}}{{{\mathrm{IC}}_{50}}^{n}{+x}^{n}})$$

### 3.3. Molecular Simulation Assays

#### 3.3.1. Molecular Docking

All ligand structures were prepared by Maestro 9.0 within the Schrödinger package. The 3D structures of all the studied compounds were created with Marvin sketch, and the initial lowest energy conformations were calculated with LigPrep. For protein preparation, the crystal structure coordinates of the human telomerase holoenzyme were obtained from the reference and then prepared with the Protein Preparation Wizard. All hydrogen atoms were added. The protein structure was then aligned with a similar protein discovered from another species to confirm the binding site (PDB:5CQG). Subsequently, the OPLS_2005 force field was used to optimize the protein energy and eliminate steric hindrance. For all dockings, the grid center was placed at the centroid of the ligand-binding site, and a 24 × 24 × 24 Å grid box size was used. All dockings were performed with Glide using the XP protocol. The docking poses were analyzed by PyMOL [28].

#### 3.3.2. MD Simulations

Compounds **5c**, **6c** and BIBR1532 were used to form three complexes with the protein refer to the molecular docking results. By using the Desmond 2014.2 software (Schrodinger, Harrison, NYC, USA) suite (D. E. Shaw Research), all-atom (OPLS3 force field) explicit water MD simulations for each researched compound were performed via Maestro MD systems and were built with the use of the highest-scoring docking pose. Specifically, the coordinate files of the human telomerase holoenzyme were obtained from the reference. An orthorhombic box with a boundary distance of 10 Å was generated to define the binding pocket. During the simulation, the Desmond default OPLS3 force field was used. All the 100 ns simulations were run on our servers [29,30]. Moreover, the RMSD, RMSF and ligand–protein interaction were monitored to determine the stability of the docking complexes.

## 4. Conclusions

In this study, twelve novel flavonoid derivatives were designed and synthesized, and ten were evaluated as potential telomerase inhibitors. It was surprising that compound **5c** showed high inhibitory activity against telomerase with an IC_50_ < 50 μM. Moreover, compound **5c** exhibited potent broad-spectrum anticancer activity against the A549, HepG2, HeLa, MGC-803 and SGC-7901 cell lines in vitro. The binding modes of BIBR1532 and compound **5c** to telomerase indicated that the conserved residues Lys 437 and Asn 421 were important for ligand binding via hydrogen bonding interactions, which could explain the SAR of these compounds and match the active in vitro data. These results help the reasonable design of the more potent telomerase inhibitors based on the structural scaffolds of natural products for cancer therapy.

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
