# Peer review of "Design, Synthesis and Molecular Docking Analysis of Flavonoid Derivatives as Potential Telomerase Inhibitors"

_molecules, 2019, doi:10.3390/molecules24173180_

Round 1

Reviewer 1 Report

The authors present very interesting and valuable material in the publication. However, some corrections are necessary:
1.
Please explain what the main reaction products were, since the yields of 6a-6c and 14a-14f are less than 50%.
2.HR MS is spelled incorrectly, with the error: HR-MS (line 214), there should be a space in the inscription HRMS (ESI) in the experimental part.
3.In line 113 the word "The most ..." is used twice.
4. The cytotoxicity of 5c to normal human cells should also be reported.
5.Ref. 1 should be changed to the statistical year presenting the global problem of cancer in the world.
6.The anticancer mechanism of action of 5c should be clearly specified in the discussion.

Author Response

Part A (to Reviewer 1)

Please explain what the main reaction products were, since the yields of 6a-6c and 14a-14f are less than 50%.

Many thanks for the reviewer’s question. The main products are our target compounds, which can be indicated by the TLC monitoring. The lower yields of the target compounds 6a-6c and 14a-14f are mainly due to the difficulty of separating products in the final step. After the perdeacetylated handling, the compounds obtained have higher polarity, which makes the material loss.

HR MS is spelled incorrectly, with the error: HR-MS (line 214), there should be a space in the inscription HRMS (ESI) in the experimental part. In line 113 the word "The most ..." is used twice.

Many thanks for the reviewer’s careful examination. We corrected them in the manuscript.

The cytotoxicity of5c to normal human cells should also be reported.

Many thanks for the reviewer’s suggestion. Actually, the cytotoxicity of compound 5c to normal human cells (Hacat: human immortalized keratinocyte cells; BEAS-2B: human bronchial epithelial cells) were also tested before. Since there are few cytotoxicity tests of flavonoid derivatives reported, we did not descript relative contents. According to the reviewer’s request, we added the data and explanation in revised manuscript (Lines 146-148 and Table 2).

1 should be changed to the statistical year presenting the global problem of cancer in the world.

The latest statistics of year 2018 was added in the revised manuscript. Many thanks for the reviewer’s suggestion.

The anticancer mechanism of action of5c should be clearly specified in the discussion.

Many thanks for the reviewer’s suggestion. The main aim of this manuscript is to introduce the novel compound as potent telomerase inhibitor, since the mechanism of action for telomerase is so complicated, up to now we have not been ready to discuss corresponding mechanism. Detailed anticancer mechanism of action of 5c will be the next goal of our further study.

Reviewer 2 Report

The manuscript entitled "Design, synthesis and molecular docking analysis of flavonoid derivatives as potential telomerase inhibitors" by Fan et al describe how the authors, using a modified TRAP assay designed and synthesized a dozen potential human telomerase inhibitors starting from the scaffolds of two series of flavonoid derivatives. One of them, called 5c, is described by the authors as having potent inhibitory activity against telomerase and anticancer activity against a panel of  tumor cell lines. They also performed a number of experiments trying  to elucidate the binding mode of  compound 5c. This finding helps the rational design of more efficient telomerase inhibitors based on the structural scaffolds of natural products.

To start with the authors in the introduction refers to the fact that most of the tumor cells have telomerase, they mentioned such finding as "recent studies." That was first described by the seminal work of Kim et al. in Science in 1994. The strong statement starting on page 38 could be largely debated. If authors think that his position is correct, they should prove it. Even worst, at line 72 authors state: " In this report, the active site of the protein structure was confirmed via alignment with a similar protein discovered from Tribolium castaneum (PDB:5CQG), whose structure is in complex with the highly specific inhibitor BIBR1532 ". Later " Due to the side effects of synthetic products, such as multi-drug resistance, toxicity, and poor bioavailability, it is preferred that telomerase inhibitors be isolated from natural plant and marine materials." BIBR1532 is a synthetic product. A contradiction in terms. On line, 71 authors said, "To date, for the human telomerase holoenzyme, there are no X-ray crystalline structures available. That statement is not rigorous; the holoenzyme of many of the components of telomerase and parts of the union of two components  are crystallized and could be found in the usual databases. On the other hand, for the components that do not have X-ray crystalline structure defined, has been proof that modeling of those components is a good way to found active telomerase inhibitors (doi: 10.3390/ijms19103216) The experimental part of the manuscript is good and well-conducted, deserving been published. However, it seems as if the authors are experts in the design and chemical production of new molecules, but they have little knowledge about the dynamics of the telomere/telomerase complex, and the literature of telomerase inhibitors.The manuscript is good enough to be published with de condition that the authors write again the concepts mentioned ut supra.

Author Response

Part B (to Reviewer 2)

To start with the authors in the introduction refers to the fact that most of the tumor cells have telomerase, they mentioned such finding as "recent studies." That was first described by the seminal work of Kim et al. in Science in 1994. The strong statement starting on page 38 could be largely debated. If authors think that his position is correct, they should prove it。

Many thanks for the reviewer’s questions. We are so sorry for our inaccurate statement and revised our expression in the manuscript.

At line 72 authors state: " In this report, the active site of the protein structure was confirmed via alignment with a similar protein discovered from Tribolium castaneum (PDB:5CQG), whose structure is in complex with the highly specific inhibitor BIBR1532 ". Later " Due to the side effects of synthetic products, such as multi-drug resistance, toxicity, and poor bioavailability, it is preferred that telomerase inhibitors be isolated from natural plant and marine materials." BIBR1532 is a synthetic product. A contradiction in terms.

Many thanks for the reviewer’s question. There is no cocrystallization of protein and natural products complexes so far, so here we have to choose the synthetic product BIBR1532 to align the active site of telomerase. The ultimate goal is to determine the active site, so that we can design and synthesize the novel telomerase inhibitors based on natural skeletons.

On line, 71 authors said, "To date, for the human telomerase holoenzyme, there are no X-ray crystalline structures available. That statement is not rigorous; the holoenzyme of many of the components of telomerase and parts of the union of two components are crystallized and could be found in the usual databases. On the other hand, for the components that do not have X-ray crystalline structure defined, has been proof that modeling of those components is a good way to found active telomerase inhibitors (doi: 3390/ijms19103216).

Many thanks for the reviewer’s suggestion. We are so sorry for our inaccurate statement. As you said, the human telomerase holoenzyme and many of the components of telomerase and parts of the union of two components are crystallized and could be found in the usual databases. Actually, we want to express that for the human telomerase holoenzyme, there is no cocrystallization of protein and ligand complexes available, except for some of the components of telomerase (such as: PDB:5UGW). So here, the active site of the protein structure was confirmed via alignment with a similar protein in complex with the highly specific inhibitor BIBR1532.

Reviewer 3 Report

In the study of Fan Z-F et al., two series of flavonoid derivatives we synthesized creating a total of twelve compounds according to the split principle. Telomerase inhibition activity in vitro and antiproliferative activity against a panel of cancer cells lines were assayed to evaluate the possibility of these compounds to function as potential telomerase inhibitors. One of the compounds, compound 5c, showed the most potent telomerase inhibitory activity and antiproliferative activity against all tested cell lines.

As there are no availableX-ray co-crystalline structures for the human telomerase holoenzyme, the authors utilized the Tribolium castaneum (PDB:5CQG) telomerase available data, which structure is in complex with the highly specific inhibitor BIBR1532 and performed molecular docking analysis to elucidate the binding mode of active compound 5c. The results showed that similar to BIBR1532, the conserved residues Lys 437 and Asn 421 were important for ligand binding via hydrogen bonding interactions, which could explain the structural-activity relationship of these compounds, which are in accordance with the in vitro activity data.

The study is well designed and presented and the data are convincing. Please find below some comment that may improve the presentation of the paper.

Paragraph 2.2.1: Please, describe in the text how the potent inhibitory activity against telomerase is observed in the gels presented. Paragraph 2.2.1: How were the IC50 values for telomerase activity were calculated? Please give the definition and the way of calculation of IC50 in the Materials & Methods relevant section. Figure 2: The actual gel images of the inhibitory effects of the compounds against telomerase by TRAP assay are provided. It would be helpful also to express the data as relative telomerase activity (% of control) normalized to the positive control as percentage after background subtraction, and present them in a separate diagram. Lines 129 & 382: Since the study involves only in vitro antiproliferative data, please correct “antitumor” to “anticancer” activity, which is more accurate. Figure 3: In Figure 3 no SD values are shown. Also, it would be more direct to plot the antiproliferative data as cell viability curves against the range of compounds’ concentration used and present the estimated IC50 values as mean ± SD in a seperate table. Again, IC50 values need to be defined and describe the way of calculation in the relevant Materials & Methods section. Figure 3: Why 5-FU was used as a positive control here and not baicalin used earlier as a telomerase inhibitor, which has a similar mode of action? Also, provide a reference for using baicalin as a reference compound for the TRAP-PCR assay. Thorough revision of English throughout the manuscript is recommended as there are typo and syntax errors.

Author Response

Part C (to Reviewer 3)

Paragraph 2.2.1: Please, describe in the text how the potent inhibitory activity against telomerase is observed in the gels presented.

Many thanks for the reviewer’s question. The relevant test methods are supplemented in the revised manuscript (See Lines 343-348).

Paragraph 2.2.1: How were the IC50 values for telomerase activity were calculated? Please give the definition and the way of calculation of IC50in the Materials & Methods relevant section.

Many thanks for the reviewer’s question. The IC50 values for telomerase activity were calculated referring to the Ref. 4. The specific calculation of IC50 is supplemented in Paragraph 3.2.2 (Lines 376-382).

Figure 2: The actual gel images of the inhibitory effects of the compounds against telomerase by TRAP assay are provided. It would be helpful also to express the data as relative telomerase activity (% of control) normalized to the positive control as percentage after background subtraction, and present them in a separate diagram.

Many thanks for the reviewer’s suggestion. We will try this method in our further study.

Lines 129 & 382: Since the study involves only in vitro antiproliferative data, please correct “antitumor” to “anticancer” activity, which is more accurate.

Many thanks for the reviewer’s suggestion. We replaced relevant statements in the manuscript.

Figure 3: In Figure 3 no SD values are shown. Also, it would be more direct to plot the antiproliferative data as cell viability curves against the range of compounds’ concentration used and present the estimated IC50values as mean ± SD in a seperate table. Again, IC50 values need to be defined and describe the way of calculation in the relevant Materials & Methods section.

Many thanks for the reviewer’s suggestion. We revised in the manuscript. The IC50 values as mean ± SD were presented in Table 1. IC50 values was defined, and the calculation of IC50 was described in Paragraph 3.2.2.

Figure 3: Why 5-FU was used as a positive control here and not baicalin used earlier as a telomerase inhibitor, which has a similar mode of action?

Many thanks for the reviewer’s question. 5-FU, as a commercial drug, was usually used as a positive control in various anticancer activity tests, including compared with the telomerase inhibitors, although it is not a telomerase inhibitor. We chose 5-FU as a positive control only for the comparison, since there is no commercial drug as telomerase inhibitors.

Also, provide a reference for using baicalin as a reference compound for the TRAP-PCR assay.

Many thanks for the reviewer’s suggestion. Here is one related reference: Ren, X., Zhang, Z., Tian, J., Wang, H., Song, G., Guo, Q., ... Jiang, G. (2017). The downregulation of c-Myc and its target gene hTERT is associated with the antiproliferative effects of baicalin on HL-60 cells. Oncology Letters, 14(6), 6833–6840. http://doi.org/10.3892/ol.2017.7039. In this paper, baicalin, as a natural flavonoid, has the antiproliferative effects via the downregulation of hTERT, meaning that baicalin can be a potential telomerase inhibitor. Howevere, we are so sorry that we have not found a reference using baicalin as a reference compound for the TRAP-PCR assay.